# First Report of Two Gymnodimines and Two Tetrodotoxin Analogues in Invertebrates from the North Atlantic Coast of Spain

**DOI:** 10.3390/md21040232

**Published:** 2023-04-05

**Authors:** Araceli E. Rossignoli, Carmen Mariño, Helena Martín, Juan Blanco

**Affiliations:** Centro de Investigacións Mariñas (CIMA), Pedras de Corón s/n, 36620 Vilanova de Arousa, Spain

**Keywords:** Gymnodimines, LC-MS/MS, new vectors, Spain, tetrodotoxins

## Abstract

Gymnodimine D (GYM D), 16-desmethyl gymnodimine D (16-desmethyl GYM D), and two tetrodotoxin analogues have been found in invertebrates obtained from the north Atlantic coast of Spain from May 2021 to October 2022. It is the first report of GYMD and 16-desmethyl GYM D in invertebrates worldwide and of the tetrodotoxin analogues, 5,6,11 trideoxy tetrodotoxin (5,6,11 trideoxy TTX) and its isomer (referred to as 5,6,11 trideoxy-epi-TTX), in the north Atlantic Coast of Spain. In this study, we also report for the first time the detection of tetrodotoxin (TTX) in three species (the cnidaria *Calliactis parasitica*, an unidentified species, and the bivalve *Tellina donacina*). The prevalence was medium for GYM D and 16-desmethyl GYM D and low for TTXs overall. The concentrations recorded were variable, with maximum values of GYM D in the bivalve *Cerastoderma edule* (8.8 μg GYM A equivalents kg^−1^), of 16-desmethyl GYM D in the bivalve *Magellana gigas* (10 μg GYM A equivalents kg^−1^) and of TTX and 5,6,11 trideoxy TTX in the cnidaria *C. parasitica* (49.7 and 233 μg TTX equivalents kg^−1^, respectively). There is very scarce information about these compounds. Therefore, the reporting of these new detections will increase the knowledge on the current incidence of marine toxins in Europe that the European Food Safety Authority (EFSA), in particular, and the scientific community, in general, have. This study also highlights the importance of analyzing toxin analogues and metabolites for effective monitoring programs and adequate health protection.

## 1. Introduction

The frequency of harmful algal blooms (HABs) produced by marine dinoflagellates has increased worldwide over the last years, with important negative impacts on public health and on the economies of the affected areas [1]. Climate change, and the consequent rise of seawater temperature and anthropogenic activity, are causing the displacement of organisms from tropical to more temperate zones, leading to the appearance of toxic species and toxins in areas where they had previously been undetected. This fact, together with the progressive discovery of new analogues of known toxins, makes it necessary to permanently monitor the coastal waters and the living resources they support in order to protect human health and to minimize damages to the ecosystem and to economic activities.

Gymnodimines (GYMs) are a group of fast-acting cyclic imine (CI) toxins initially identified in oysters collected from New Zealand [2,3]. The mechanism of action of GYMs is the same as that of other cyclic imines, such as spirolides (SPXs), and has been described by Munday et al. [4]. These compounds interact and block muscle-type nicotinic acetylcholine receptors acting at the level of the skeletal neuromuscular junction, causing inhibition of muscle action potential. So far, eight GYMs have been identified [5] and, while the toxicity of GYM A is known [4], nearly no information relative to other analogues exists. GYM A, B, and C have been linked to the presence of the dinoflagellate *Karenia selliformis* (previously cited as *Gymnodinium*) [6,7]) in New Zealand and of *Alexandrium ostenfeldii* in the Baltic Sea (Finland) [8]. Other analogues, such as 12-methyl GYM A, 12-methyl GYM B, GYM D, 16-desmethyl GYM D, and GYM E, have been related to *A. ostenfeldii* [5,9,10,11,12] in the Netherlands, North Carolina, and the Baltic Sea.

No regulatory limit for GYMs in shellfish has yet been set in Europe or any other area of the world. There are only some proposed tentative limits for two other groups of CIs (SPXs) [13] and pinnatoxin G (PnTXG) [14]. The European Food Safety Authority (EFSA) requires additional data to assess the real risk of CIs (including GYMs) and encourages their obtention [14]. 

Tetrodotoxin (TTX) is a potent natural marine neurotoxin which blocks the sodium channels in motor nerves [15]. As a consequence, it causes skeletal muscle paralysis and death. TTX was discovered in 1909 from the ovaries of the puffer fish [16], a fish that belongs to the family of *Tetraodontidae* from which the toxin name was derived. This compound is considered the most lethal toxin found in the marine environment [17,18]. At least 30 structural analogues, with different degrees of toxicity depending on their structure [16], have been described to date (Figure 1). The origin of TTX is believed to be bacteria, such as *Pseudomonas*, *Pseudoalteromonas*, and *Vibrio*, but, occasionally, it has been reported in *Actinobacteria*, *Bacteroides*, *Firmicutes*, or *Proteobacteria* [17,19].

Only one European Union (EU) regulation, published in 2004, prevents fish species reported as TTX-bearers (*Tetraodontidae*, *Canthigasteridae*, *Molidae*, and *Diodontidae*) from market placement [20]. Some years ago, the European Commission asked the European Food Safety Authority (EFSA) for a scientific opinion on the risks to public health related to the presence of TTX and TTX analogues in marine bivalves and gastropods. In 2017, EFSA indicated that a concentration below 44 μg TTX equivalents kg^−1^ in shellfish meat could be considered not to result in adverse effects in humans. However, they also noted that more occurrence data on TTX and its analogues in edible parts of marine bivalves and gastropods from different EU waters are needed to provide a more reliable exposure assessment [21].

Galicia, in the northwest of Spain, has approximately 1200 km of coastline and is the main bivalve (particularly mussel) producer in Europe, with an average annual production of more than 238,000 metric tons per year since 2003 [22]. In the last decade, in addition to the traditional bivalve resource’s exploitation, other groups of marine organisms, such as cnidaria, algae, polychaetes, and echinoderms, have started to be harvested and commercialized.

In the present work, an exhaustive screening of the presence/absence of a high number of emerging toxins (including GYMs and TTXs) in different classes of marine invertebrates along the northwest Atlantic coast of Spain (Galicia) was carried out. This paper reports for the first time: (1) the detection of GYM D and 16-desmethyl GYM D in invertebrates in the world, (2) the appearance of TTX in some species in which it had never been reported, and (3) the presence of 5,6,11 trideoxyTTX and its isomer (probably an epimer 5,6,11 trideoxy-epi-TTX) in invertebrates in Galicia.

This report provides the new occurrence data of toxins in marine organisms that the EFSA are demanding from different EU Member States and that contribute to a better assessment of the risk that emerging marine toxins pose to human health.

## 2. Results

### 2.1. GYMs Detection

Two GYMs were detected in the analyzed samples: 16-desmethyl GYM D and GYM D (Figure 2). 

16-desmethyl GYM D (Figure 3) was present in 16 gastropod specimens (5 *Patella* spp., 1 *Nassarius* sp., 1 *Nassarius reticulatus*, 1 *Monodonta* sp., 1 *Gibbula* sp., 2 *Nucella* spp., 2 *Doris verrucosa*, 2 *Littorina* spp., and 1 unidentified species); 35 bivalves (5 *Ruditapes decussatus*, 4 *R. philippinarum*, 5 *Cerastoderma edule*, 1 *Magellana gigas*, 3 *Ostrea edulis*, 16 *Mytilus galloprovincialis*, and 1 *Aequipecten opercularis*); 2 echinoderms (*Asterina gibbosa* and *Paracentrotus lividus*); 6 crustaceans (*Carcinus maenas*, *Necora puber*, *Polybius* sp., *Balanus* sp., *Liocarcinus corrugatus*, and *L. arcuatus*); and 3 cnidaria (*Calliactis parasitica* and 2 unidentified species). These samples were obtained from numerous sampling points along the Galician geography and collected at different times of the year. Only in two cases of *Patella* spp. (from Pont V—Placeres, Pontevedra, 2 March 2022 and Vilanova de Arousa—Corón, Arousa, 14 June 2022) and one *M. gigas* (from Camariñas—Enseada Vasa, Camariñas, 19 July 2022), the concentrations were quantifiable (above LOQ), reaching, in the case of the oyster, a maximum estimated value of 10 μg GYM A equivalents kg^−1^.

GYM D (Figure 4) was present in the same samples in which 16-desmethyl GYM D was detected but in much lower concentrations. There was only one exception: a cockle (*C. edule*) from Pasaxe, Sta. Cristina (A Coruña, 1 February 2022), in which the concentrations of GYM D were quantifiable (8.8 μg GYM A equivalents kg^−1^) and higher than that of 16-desmethyl GYM D in the same sample.

### 2.2. TTXs Detection

TTX was detected in three samples: two cnidaria (*C. parasitica* from Bohído, Arousa, 24 May 2022 and an unidentified species from Ramallosa, Vigo, 5 November 2021) and one bivalve, *Tellina donacina*, from Bohído, Arousa, on 1 September 2021 (Figure 5). Only the toxin concentration in the cnidaria *C. parasitica* could be quantified (49.7 μg TTX equivalents kg^−1^), with a level which exceeded the recommended maximum legal concentration of 44 μg TTX equivalents kg^−1^ TTX.

Some TTX analogues, 5,6,11 trideoxy TTX and an isomer of 5,6,11 trideoxy TTX, with an identical fragmentation spectrum (referred hereafter as 5,6,11 trideoxy-epi-TTX) (Figure 6) were also detected in certain species of invertebrates, and, for the first time, in Galicia. These compounds appeared mainly in the Bohído area (Ría de Arousa), where they were detected on 24 May 2022 in the crustacean *L. corrugatus*, in the gastropod *N. reticulatus*, in the cnidaria *C. parasitica*, and in the echinoderm *A. gibbosa*, and also on 29 July of the same year in the polyplacophore *C. angulata*. The presence of these compounds was also observed in an unidentified cnidaria species collected in Porto do Son, Arnela (Muros-Noia), on 4 March 2022. The concentrations present in the samples varied remarkably from below the LOQ (in the crustacean, the gastropod, and the echinoderm) to 118 and 233 μg TTX-equivalents kg^−1^ in the two cnidaria, respectively.

## 3. Discussion

This is the first report of the detection of GYM D and 16-desmethyl GYM D in invertebrates worldwide. There is very little information published about these compounds. The fragmentation spectra found coincide perfectly with those reported by Harju et al. [11], Martens et al. [12], and Zurhelle et al. [5] for these compounds in *A. ostenfeldii*. Up to now, the only GYMs found in shellfish have been GYM A in New Zealand [2], Tunisia [23,24], South Africa [25], China [26], Italy [27], France [28], and Spain [29]; GYM B just in Tunisia [23,24]; and supposed GYM G, GYM H, GYM I, GYM J, GYM F, 5 isobaric analogues of GYM B/C, and some ester metabolites of GYMs in a pool of Mediterranean samples [30]. As far as we know, there is no report of the presence of GYM D or 16-desmethyl GYM D in shellfish. The detection of the GYM D and the 16-desmethyl GYM D in different classes of invertebrates, including bivalves, gastropods, echinoderms, crustaceans, and cnidaria, indicates that the prevalence in the area is noticeable. The species responsible for the production of GYMs in the area has not been identified yet. *A. ostenfeldii* and *Karenia selliformis* have been shown to produce GYMs. 

*A. ostenfeldii* can produce 16-desmethyl GYM D and GYM E [5,12], GYM A, 12-methyl GYM A [9,10], and more than 30 other, albeit minor, GYM-like compounds [11], indicating that a considerable diversity of GYMs might be a common feature for this species. *K. selliformis* has only been found to produce GYMs A, B, and C [2,31,32]. Up to now, in Galicia, the dinoflagellate that is supposed to be the main GYMs producer, *A. ostenfeldii*, has not been unequivocally identified yet, so other possible origins of this compound cannot be ruled out. Nevertheless, the fact that *A. ostenfeldii* is the only species know to produce GYMs of the group D points to this species as the most likely responsible for the production of these toxins in Galicia.

The detection of the toxins throughout different seasons, as well as in many sampling areas, prevents the identification of any specific spatio-temporal pattern. 

The 16-desmethyl GYM D concentrations observed were, in most cases, lower than the LOQ of the method. Only two limpets (*Patella* spp. from Ría de Pontevedra and Arousa) and one oyster (*M. gigas* from Ría de Camariñas), in which a maximum estimated value of 10 μg GYM A equivalents kg^−1^ was recorded, had quantifiable concentrations. GYM D could also be detected in the same samples, although at much lower concentrations than the 16-desmethyl GYM D. There was only one exception with a cockle (*C. edule*) from Pasaxe, Sta. Cristina (A Coruña, 1 February 2022), in which the concentrations of GYM D were quantifiable (8.8 μg GYM A equivalents kg^−1^) and higher than those of the 16-desmethyl GYM D. The presence of both GYMs in the same samples suggests that 16-desmethyl GYM D could be a product of the biotransformation of GYM D.

The information about these compounds is very scarce and nothing is known about their toxicities. Hence, no conclusions can be drawn about the real risk that the concentrations found in this study may pose to the population. 

In this study, we also report for the first time the detection of TTX in three species (the cnidaria *C. parasitica*, an unidentified species, and the bivalve *T. donacina*) and the presence of the TTX analogues 5,6,11 trideoxy TTX and 5,6,11 trideoxy-epi-TTX in invertebrates from Galicia. TTXs have been found in different species of bivalves, gastropods, other molluscs, echinoderms, crustaceans, and even flatworms from different countries (reviewed in several articles [33,34]) all over the world. In the Atlantic coast of the Iberian Peninsula, the presence of TTX has only been documented in Galicia (Spain) in one sample of cockle (2.3 µg kg^−1^) and one sample of an oyster (below LOQ) (species not reported), without analogues [35], and in Portugal in some species of gastropods [36,37,38,39]. TTX was also reported from insular Portugal (São Miguel Island, Azores) in the pufferfish *Sphoeroides marmoratus* and the echinoderm *Ophidiaster ophidianus* [40]. The three positive samples found in this study came from two different areas (Bohído, Ría de Arousa and Ramallosa, Ría de Vigo) and were collected in different months and years. Only in the case of the cnidaria (*C. parasitica*), the obtained concentration was quantifiable (49.7 μg TTX equivalents kg^−1^). This is the highest concentration of TTX found to date in invertebrates in Spain, and it slightly exceeds the recommended maximum legal concentration of 44 μg TTX-equivalents kg^−1^ [21]. Although most of the TTX poisonings reported across the world are related to pufferfish consumption, several other marine organisms, such as the blue-ringed octopus, gobies, horseshoe crab, starfish, and gastropods, may also contain TTX, and their consumption may result in poisoning and death [41]. The data obtained in this study expand the range of species which can be vectors of TTX and, consequently, whose consumption may pose a risk to human health. 

Finally, in the present study, we also detected the analogues 5,6,11 trideoxy TTX and its isomer, probably an epimer (5,6,11 trideoxy-epi-TTX), in different species of crustaceans, gastropods, cnidaria, and echinoderms. Most positive samples for these analogues (5 out of 6) were collected in the same area (Arousa-Bohído, Ría de Arousa) between May and July 2022. This fact suggests that this area and time of the year are hotspots for TTX analogues. A complete review of the presence of these analogues in European waters, with a special emphasis on marine bivalves and gastropods, is summarized in Katikou [42]. Only in the case of the cnidaria *C. parasitica*, TTX was also present in the sample, but in an amount 4.5 times lower than that of 5,6,11 trideoxy TTX, which is similar to that detected by Rodríguez et al. [37] in *Charonia lampas*. However, in the review published in 2019 about recent advances after the EFSA Scientific Opinion by Katikou [42], it was pointed out that, in the samples in which the analogues were present, the dominant toxin was TTX, representing 90% of the total TTX concentration, on average. She also indicated that no species-related differences were noted in the occurrence of 4-epiTTX, 5,6,11 trideoxyTTX, and 4,9-anhydroTTX detected in the analysed clams, oysters, and mussels with mean proportions of 7%, 29%, and 7%, respectively [42]. However, it is evident that, for almost all other species, such differences between TTX analogues must exist. Here, we show that, contrarily to what was pointed out by Turner et al. [43] for bivalve molluscs, shellfish can contain only TTX analogues, without the parent toxin being present at a detectable concentration. In this study, the 5,6,11 trideoxy TTX concentrations quantified in the invertebrates were variable, measuring from below the LOQ (in the crustacean, gastropod, and the echinoderm) to 118 and 233 μg TTX equivalents kg^−1^ in the two cnidaria studied. 

Comparisons of the concentrations of TTX and 5,6,11 trideoxy TTX obtained by LC-MS, with their toxicities estimated by mouse bioassay, demonstrated that the analogue is almost not toxic [37]. This is probably due to a lower affinity for binding to the sodium channel as a consequence of a smaller number of hydroxyl groups than TTX [16]. Taking this into account, the concentrations reported in this study should not pose any risk to public health. However, some studies suggest a possible metabolic pathway through the oxidation of 5,6,11 trideoxy TTX to TTX [42,44,45] and some other complex transformations, including dehydrogenation of the C-10-hydroxyl, which is believed to be essential for the affinity for intracellular targets [37] and that can, in consequence, increase the toxicity of the compound. 

The detection of new toxin analogues in invertebrates or the toxins in new vectors, such as those in our study, highlight the importance of analysing toxin analogues and metabolites for effective monitoring programs and adequate health protection.

## 4. Materials and Methods

### 4.1. Chemicals and Solvents

The chemicals and reagents used were of analytical or higher grade. Acetonitrile (MeCN, LC-MS grade) and methanol (MeOH, HPLC grade) were purchased from Scharlab (Barcelona, Spain) and VWR (Barcelona, Spain), respectively. Ultrapure water was obtained from a Milli-Q Gradient system fed with an Elix Advantage-10 (Millipore Iberica, Madrid, Spain). Ammonium hydroxide (NH_4_OH, 25%), sodium hydroxide (NaOH > 99%), formic acid (98–100%), and glacial acetic acid (AcOH) were obtained from Merck (Barcelona, Spain), and hydrochloric acid (HCl, 37%) was obtained from Panreac (Barcelona, Spain). The graphitized carbon Supelclean ENVICarb (250 mg/3 mL) cartridges were acquired from Supelco (Bellefonte, PA, USA). 

Certified reference standard (CRM) for GYM A was acquired from the Institute for Marine Biosciences, National Research Council (NRC) (Halifax, NC, Canada) and for TTX from Cifga S.A. (Lugo, Spain). Working standard solutions were prepared by diluting the stock solution with 50% methanol for GYM A or acetic acid 0.1 mM for TTX. 

### 4.2. Sampling

Different species of marine invertebrates were collected along the Galician Atlantic coast (northwest Spain, Figure 7) from May 2021 to October 2022 with an approximately monthly frequency, complemented with some opportunity sampling. Samples were collected from the intertidal zone at low tide manually or from the subtidal zone by freediving or dredging. A complete table including date, sampling location, Ría, class, species, and type of analysed toxin is reported in the Appendix A.

For GYMs screening, a total of 437 sample analyses were performed. In total, 271 corresponded to bivalves, 21 to cnidarias, 30 to crustaceans, 15 to echinoderms, 92 to gastropods, 4 to polychaetes, 1 to poriferous, and 3 to sea squirts.

In the case of the TTX study, a total of 785 samples distributed across bivalves (498), cnidarias (32), crustaceans (44), echinoderms (26), gastropods (175), polychaetes (8), and poriferous (2) were extracted and analyzed.

Finally, for 5,6,11 trideoxy TTX, the 20 samples analyzed were distributed across 1 bivalve, 3 cnidarias, 9 crustaceans, 4 echinoderms, 2 gastropods, and 1 polychaete. Once collected, the samples were transferred to the laboratory in portable refrigerators and frozen at −20 °C until their extraction.

### 4.3. Toxin Extraction and LC-MS/MS Analysis

#### 4.3.1. GYMs

Once the samples were thawed, the soft tissues were homogenized in methanol 100% (1:4 *w*/*v*) with an Ultraturrax T25 (IKA, Staufen, Germany) to extract the toxins. The obtained extracts were clarified by centrifugation at 19,000× *g* and by filtering through 0.22 µm polyethersulphone PES syringe filters (Membrane Solutions from Jasco Analítica, Madrid, Spain) and were finally analyzed using liquid chromatography coupled to mass spectrometry (LC-MS/MS). The analyses were carried out on an Exion LC AD™System (SCIEX, Framingham, MA, USA) coupled to a Qtrap 6500+ mass spectrometer (SCIEX) through an IonDrive TurboV interface in electrospray mode. A total of 13 GYMs were screened according to Rossignoli et al. [46], with slight modifications. Briefly, the toxins were separated using a Gemini NX C18 column 50 mm (length) × 2 mm (id), 3 µm (particle size), from Phenomenex (Torrance, CA, USA). Mobile phase A was water and phase B was MeCN 90%, both containing 6.7 mM NH_4_OH (pH = 11). The gradient started with 22% B, was maintained for 0.5 min, followed by a linear increment to reach 95% B at minute 3.85, and maintained until minute 6.25. The composition of the mobile phase was then returned linearly to the initial composition in 0.5 min and maintained for 2 min before the next injection. The flow rate was 0.4 mL min^−1^, the injection volume was 2 µL, and the column temperature was 40 °C. Limit of detection (LOD, s/*n* = 3) and limit of quantification (LOQ, s/*n* = 10) for GYMA were 1.2 and 3.9 µg kg^−1^, respectively. The mass spectrometer parameters were set to ion source gas 1, 75 (arbitrary units); ion source gas 2, 75 (arbitrary units); ion spray voltage, 5000 (positive); capillary temperature, 600 (°C); curtain gas, 30; and collision gas, medium. Declustering potential (DEP), entrance potential (EP), and collision cell exit potential (CXP) were ±80 v, 15 v, and ±11 v, respectively. The transitions with the collision energies used for the screening of GYMs are shown in Table 1. The absence of certified reference material for 16-desmethyl GYM D forced us to quantify it as a GYM A equivalent by an external standard method using dilutions of GYM A solution and assuming an equimolar response between both compounds. [12,47,48,49]. 

#### 4.3.2. TTXs

For TTXs screening, samples were extracted according to Boundy et al. [50]. Briefly, a five-gram sample was weighed into a 50 mL polypropylene centrifuge tube, mixed with 5 mL of acetic acid 1% on a vortex for 90 s, and placed in a boiling water bath for 5 min. After cooling by placing the tube in an ice bath, the sample was remixed on a vortex mixer for 90 s and centrifuged at 4000× *g* for 10 min at 4 °C. An aliquot (1 mL) of the supernatant was mixed with 5 µL of ammonium hydroxide (25% NH_4_OH), and 400 µL of the mixture was loaded onto a graphitized carbon cartridge Supelclean ENVICarb (250 mg/3 mL) conditioned with 3 mL of 20% MeCN plus 1% AcOH, followed by 3 mL of 0.025% NH_4_OH. Cartridges were washed with 700 µL of Milli-Q water and eluted with 2 mL of 20%MeCN plus 1% AcOH. Finally, 100 µL of the obtained eluate was diluted with 300 µL of MeCN into a propylene vial and analyzed using LC-MS/MS.

The LC-MS/MS equipment and the mass spectrometer parameters were the same as in the previous Section 4.3.1. In this case, TTX toxins were screened using a Waters Acquity UPLC Glycan BEH Amide 2.1 mm × 150 mm, 1.7 µm, under alkaline conditions, according to Boundy et al. [50]. Mobile phases were A, water/formic acid/NH_4_OH (500/0.075/0.3 *v*/*v*/*v*), and B, acetonitrile/water/formic acid (700/300/0.2 *v*/*v*/*v*). The gradient started at 0.4 mL min^−1^ with a proportion 4:96 A:B that was maintained for 3.5 min. Then, a linear change was started, reaching 50:50 A:B at minute 6, maintaining this composition until minute 7.5 but increasing the flow rate to 0.5 mL min^−1^. The composition was then returned linearly to the initial composition (96% B) in 0.5 min at 0.5 mL min^−1^ and maintained for 1.5 min at the same conditions. The flow rate was increased again up to 0.8 mL min^−1^ in minute 9.8 and maintained for 0.6 min. Finally, the flow was returned to 0.4 mL min^−1^ over 0.5 min and maintained for 0.1 min before the next injection. The injection volume was 1 µL, and the column temperature was 60 °C. The total run took 11 min. 

LOD and LOQ for TTX were established at 0.18 and 0.6 µg kg^−1^, respectively. The transitions with their corresponding collision energies used for the screening of TTXs are shown in Table 2. TTXs in the extracts were quantified by an external standard method using dilutions of a TTX solution in 0.1 mM AcOH and assuming an equal response of the analogues on a molar basis [47,48,49]. 

## Figures and Tables

**Figure 1 marinedrugs-21-00232-f001:**
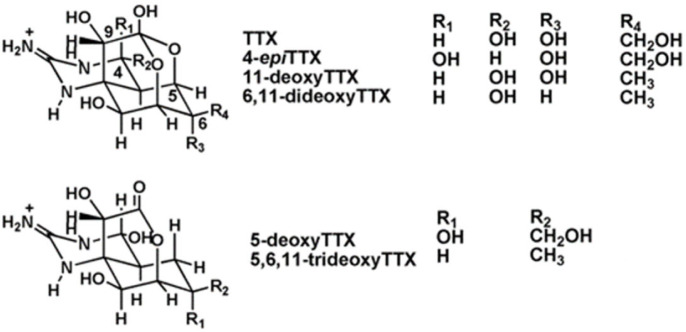
Chemical structure of TTX and its main analogues.

**Figure 2 marinedrugs-21-00232-f002:**
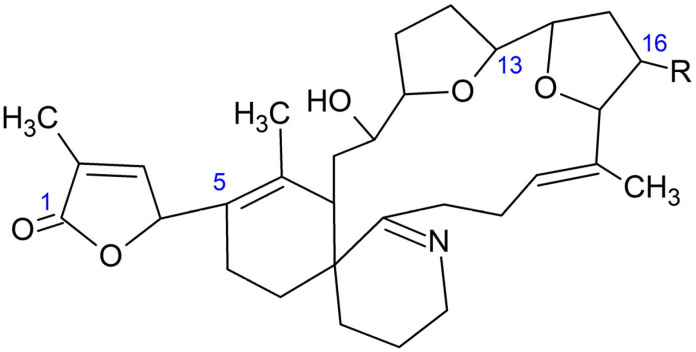
Structure of 16-desmethyl GYM D (R = H) and GYM D (R = CH_3_).

**Figure 3 marinedrugs-21-00232-f003:**
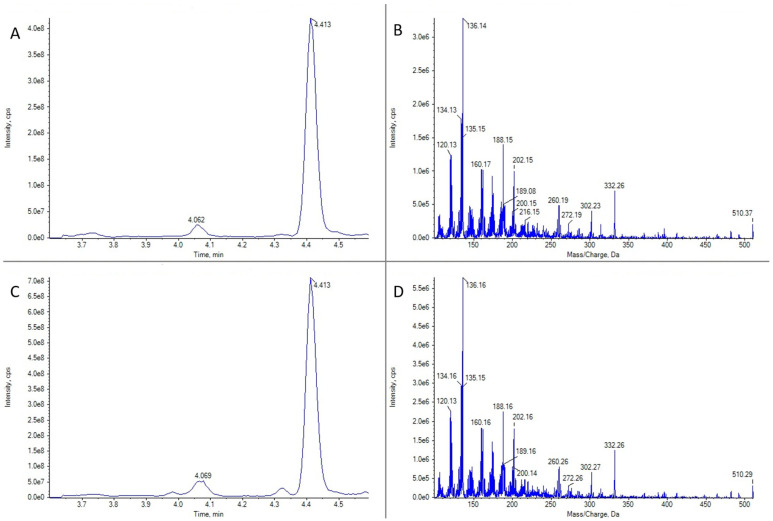
LC-MS/MS analysis in EPI mode for 16-desmethyl GYM D for two limpet samples of *Patella* spp. (**A**) Ion chromatogram for *m*/*z* 510.3 in limpet sample 1. (**B**) LC-MS/MS mass spectrum of the highest peak in panel (**A**). (**C**) Ion chromatogram for *m*/*z* 510.3 in limpet sample 2. (**D**) LC-MS/MS mass spectrum of the highest peak in panel (**C**).

**Figure 4 marinedrugs-21-00232-f004:**
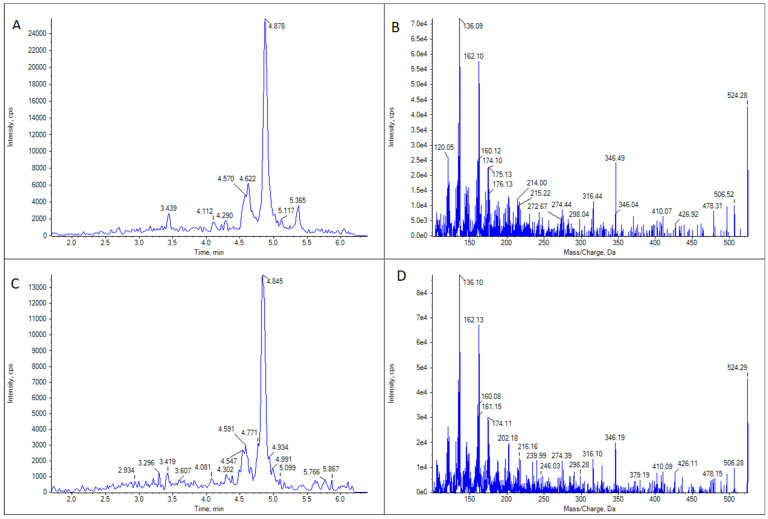
LC-MS/MS analysis of GYM D in two limpet (*Patella* spp.) samples. (**A**) Extracted ion chromatogram for the transition *m*/*z* 524.3 > 136.2 in limpet sample 1. (**B**) CID fragmentation spectrum of the highest peak in panel (**A**). (**C**) Extracted ion chromatogram for the transition *m*/*z* 524.3 > 136.2 in limpet sample 2. (**D**) CID fragmentation spectrum of the highest peak in panel (**C**).

**Figure 5 marinedrugs-21-00232-f005:**
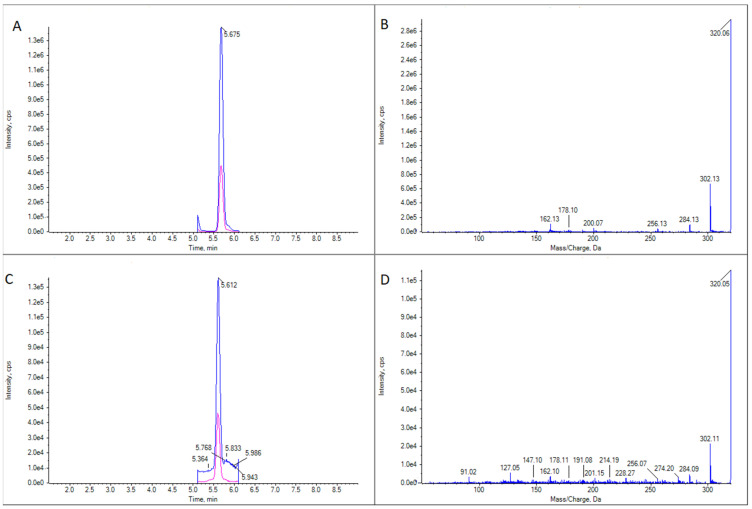
LC-MS/MS analysis of a TTX standard and a sample of *Calliactis parasitica*. (**A**) Extracted ion chromatogram for the transitions *m*/*z* 320.1 > 302.1 and 320.1 > 162.1 in a TTX standard. (**B**) CID fragmentation spectrum for the peak in panel (**A**). (**C**) As panel A for a sample of *C. parasitica*. (**D**) As panel (**B**) for the peak in panel (**C**).

**Figure 6 marinedrugs-21-00232-f006:**
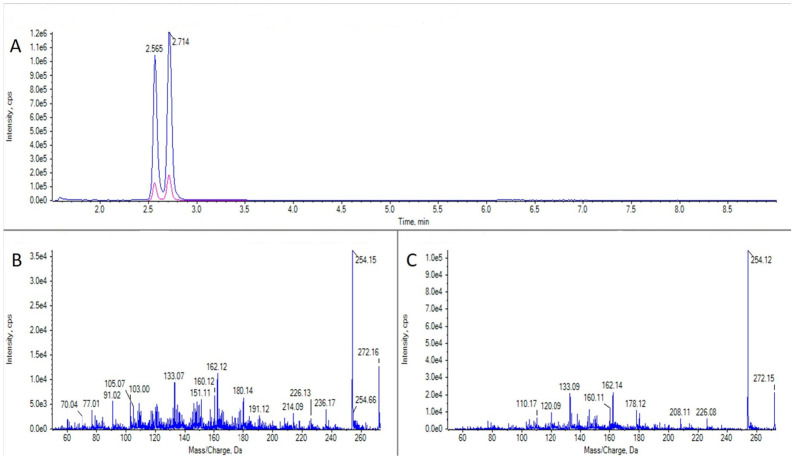
LC-MS/MS analysis of 5,6,11 trideoxy TTX from a sample of *Calliactis parasitica*. (**A**) Extracted ion chromatogram for the transitions *m*/*z* 272.1 > 254.1 and 272.1 > 162.1 in a sample of *C. parasitica.* (**B**) CID fragmentation spectrum of the first peak in panel (**A**) (5,6,11 trideoxy-epi-TTX). (**C**) CID fragmentation spectrum of the second peak in panel (**A**) (5,6,11 trideoxy TTX).

**Figure 7 marinedrugs-21-00232-f007:**
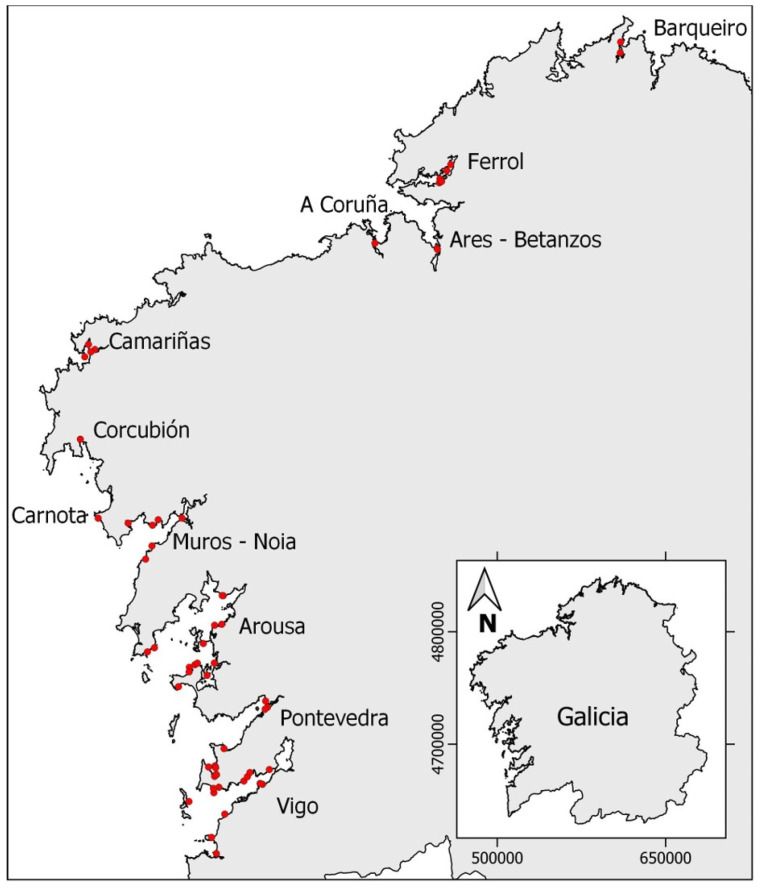
Sampling locations on the Galician coast. Red points indicate the approximate sampling point in each Ría. The inserted map shows the total autonomous community of Galicia.

**Table 1 marinedrugs-21-00232-t001:** MS/MS fragmentation parameters for GYMs screening. ESI = electrospray ionization mode, Q1 = *m*/*z* ratio in the first quadrupole, Q3 = *m*/*z* ratio in the third quadrupole, and CE = Collision energy.

Toxin	ESI	Q1	Q3	CE (v)
GYM A	POS	508.3	392.2	60
GYM A 2	POS	508.3	136.2	60
GYM B, C, D	POS	524.3	136.2	60
GYM B, C 2	POS	524.3	320.2	60
GYM D 2	POS	524.3	346.2	60
GYM E, F	POS	526.3	136.2	60
GYM E 2	POS	526.3	348.2	60
GYM F 2	POS	526.3	392.2	60
GYM G, H	POS	556.3	136.2	60
GYM G 2	POS	556.3	304.2	60
GYM H 2	POS	556.3	320.2	60
GYM I	POS	540.3	136.2	60
GYM I 2	POS	540.3	320.2	60
12-methyl GYM A, GYM J	POS	522.3	136.2	60
12-methyl GYM A 2	POS	522.3	406.2	60
GYM J 2	POS	522.3	424.3	60
16-desmethyl GYM D	POS	510.3	136.2	60
16-desmethyl GYM D 2	POS	510.3	332.2	60

**Table 2 marinedrugs-21-00232-t002:** MS/MS fragmentation conditions for TTX screening. ESI = electrospray ionization mode, Q1 = *m*/*z* ratio in the first quadrupole, Q3 = *m*/*z* ratio in the third quadrupole, and CE = Collision energy.

Toxin	ESI	Q1	Q3	CE (v)
TTX/4-epi TTX	POS	320.1	302.1	31
TTX/4-epi TTX	POS	320.1	162.1	40
4-9 anhydro TTX	POS	302.1	162.1	40
4-9 anhydro TTX	POS	302.1	256.1	31
11 deoxy TTX	POS	304.1	286.1	31
11 deoxy TTX	POS	30.4	176.1	40
5 deoxy TTX	POS	304.1	286.1	31
5 deoxy TTX	POS	304.1	176.1	40
5,11/6,11 dideoxy TTX	POS	288.1	270.2	31
5,11/6,11 dideoxy TTX	POS	288.1	224.2	40
11-norTTX-6-ol	POS	290.1	272.2	31
11-norTTX-6-ol	POS	290.1	162.1	40
5,6,11-trideoxy TTX	POS	272.1	254.1	31
5,6,11-trideoxy TTX	POS	272.1	162.1	40

## Data Availability

Data are available on request from the Centro de Investigacións Mariñas.

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
