# Peer review of "First Report of Two Gymnodimines and Two Tetrodotoxin Analogues in Invertebrates from the North Atlantic Coast of Spain"

_marinedrugs, 2023, doi:10.3390/md21040232_

Round 1

Reviewer 1 Report

The manuscript entitled ‘‘First report of 16-desmethyl gymnodimine D and two tetrodotoxin analogues in invertebrates from the north Atlantic Coast of Spain” as an article of Marine Drugs (Manuscript ID: marinedrugs-2253671) by Rossignoli and coauthors describes 16-desmethyl gymnodimine D and TTXs were attempted to detect in a variety of marine invertebrates in Galicia and were detected in several species. Although the authors have been vigorously investigating a variety of marine invertebrates and obtained unique results, there are several critical issues in this paper that need to be resolved before it can be considered for publication.

Major comments:

1. You state that you have detected the epimer of 5,6,11-trideoxyTTX as a new TTX analog in invertebrates, on what basis did you identify the epimer? I have not found any reports of 4epi-5,6,11-trideoxyTTX being identified to date.

2. The scientific evaluation of this paper will be made by the readers, but it lacks the basic information necessary to do so. For example, although the study is conducted on a wide variety of marine invertebrates, the specific species names are limited to those species for which the substance of interest was detected in this study. Some of the same species may possess the substance in different areas of the ocean, while others may not (in fact, you were finding it, weren't you?). Although not required in the text, a list of species examined in this study should be included as supplemental data, including information such as time of collection, location (this can be rough if species protection, etc. is needed), and size.

3. Figures in scientific papers should not use screen captures from the control PC monitor of the analytical instrument. They should be simple and clear (Figures 3 – 5); in Figure 2, the selected elution times are misaligned between the standard and the sample, making them difficult to view.

4. Figure 2 is a copy and paste of the figure from the cited paper, which is a serious infraction of academic ethics. In the first place, this paper is not a Review, so it should be sufficient to show only the substance to be studied.

Specific comments

Figure 1: Since this paper is not a Review, it is not necessary to show all GYMs variants. It is sufficient to know the structure of the 16-desmethyl gymnodimine D that are the primary focus of this study (it might be better to include the 16-desmethyl gymnodimine D in the Results section). Also, the paper from which this figure is taken did not include any description of the structure of GYMs.

Line 64: UE -> EU?

Line 92: 18 gastropods -> 18 gastropod specimens?

Line 201: TTXs analogues -> “TTXs” or “TTX analogues”?

Line 248: frozen -> What is the temperature?

Figure 6: It should be possible to indicate which part of Galicia this map is an enlargement of.

Line 300: 0.5 mL mL-1 -> 0.5 mL min-1?

Table 2: nor-TTX -> 11-norTTX-6,6-diol, 11-norTTX-6(S)-ol, or 11-norTTX-6(R)-ol?

Reviewer 2 Report

My comments are included within pdf document, this research shows the presence of 16-desmethyl GYM D and analogs of TTX in different species of invertebrates, however results of most part of samples were not were shown, to know and understand if the negative results has relationship with the sampling period or when GYM D and analogs of TTX were detected in same site,  both groups of toxins were found in analyzed species, in although trace concentrations

Reviewer 3 Report

Dear Editor Greetings.

I hereby inform you that I have read in detail the article entitled “First report of 16-desmethyl gymnodimine D and two tetrodotoxin analogues in invertebrates from the north Atlantic  Coast of Spain” by  Rossignoli et al.

The article is very well written and presents the first-time detection of 16-desmethyl gymnodimine D (16-desmethyl GYM D) and two tetrodotoxin analogues  in invertebrates obtained from the north Atlantic coast of Spain.

Therefore, it represents a great contribution to increase the knowledge that the European Food Safety Authority (EFSA), in particular, and the scientific community, in general, have about the current incidence of marine toxins in Europe.

Below are my suggestions:

Line 25 Keywords: I suggest changing the following keywords= Analogues and first detection, for others more appropriate for the article.

Introduction

Line 37- 51 Please include the possible effects produced by Gymnodimines (GYMs) to adequately justify the interest of the analysis and why the information generated is important.

Please improve the quality of Figures 1 and 2.

Results:

Are the detected analogs of Gymnodimines (GYMs) produced by the dinoflagellate or are they the product of bioconversion?

Materials and Methods

When referring to 801 samples,

were 801 analyses performed?

Were 801 Toxin Extraction performed?

4.3. Toxin Extraction and LC-MS/MS analysis

Please include the LOD and LOQ for the corresponding analyses.

In addition, include that eventually non-toxic/toxic material was disposed of according to biosafety protocols.

Round 2

Reviewer 1 Report

The manuscript entitled ‘‘First report of 16-desmethyl gymnodimine D and two tetrodotoxin analogues in invertebrates from the north Atlantic Coast of Spain” as an article of Marine Drugs (Manuscript ID: marinedrugs-2253671) by Rossignoli and coauthors has been revised and improved in the areas I have pointed out. However, there are several problems with the paper that need to be resolved before it can be considered for publication.

Major comments:

1. I understand that Toxins (as well as Marine Drugs) are CCBY compliant. However, the first manuscript was completely copy-pasted with the entire figure in question. The idea that copying and pasting of figures is acceptable when copying and pasting of text is not (even if a citation is shown) raises questions about the authors' attitude toward their research. However, we think that the value of this paper has increased greatly because the new version has resolved the issue of the figure.

2. I understand that Figures 3-5 used direct exports from the analysis software. However, the graphs in these figures also show text and other items that are not considered necessary to explain the contents of this figure. While the essential parts of the figures should not be processed, I think it is problematic to display parts of the figures that are not necessary for the argument. In general, journals are required to present only the necessary parts in a simple and clear manner, and Marine Drugs is no different. Haven’t you presented your papers that way in the past?

Specific comments

Figures 3-5: In addition to the points noted above, what does the blue arrows on the left side of the vertical axis indicate? If this needs to be shown, you should explain what this indicates in the caption.

Figure 3: In addition to the points noted above, a gray line BOX is drawn on A and C. What do these indicate?

Figure 4: A-D are not indicated in the figure.

Round 3

Reviewer 1 Report

All the areas I mentioned in the previous round were improved. I think this version is acceptable for publication.